# The Leisure Time Physical Activity Questionnaire for People with Disabilities: Validity and Reliability

Cameron M. Gee [1,2] , Ava Neely [3], Aleksandra Jevdjevic [3], Kenedy Olsen [3] and Kathleen A. Martin Ginis [1,3,4,5,*]

1 International Collaboration on Repair Discoveries, University of British Columbia, Vancouver, BC V5Z 1M9, Canada; cgee@icord.org
2 Department of Orthopaedics, University of British Columbia, Vancouver, BC V5Z 1M9, Canada
3 School of Health and Exercise Sciences, University of British Columbia, Kelowna, BC V1V 1V7, Canada; neely.ava@gmail.com (A.N.); aleks.jevdjevic@ubc.ca (A.J.); kenedy.olsen@mail.utoronto.ca (K.O.)
4 Department of Medicine, Division of Physical Medicine & Rehabilitation, University of British Columbia, Vancouver, BC V5Z 2G9, Canada
5 Centre for Chronic Disease Prevention and Management, University of British Columbia, Kelowna, BC V1V 1V7, Canada
* Correspondence: kathleen_martin.ginis@ubc.ca

**Abstract:** There is an urgent need for valid and reliable measures of physical activity (PA) participation for use among people with physical and/or sensory disabilities. This study involved adapting the Leisure Time PA Questionnaire for People with Spinal Cord Injury for use in individuals with disabilities (i.e., the LTPAQ-D) and performing a preliminary evaluation of its content validity, construct validity, and same-day test–retest reliability in people with disabilities. User interviews assessed the content validity ($n = 5$). A cross-sectional study assessed the construct validity and same-day test–retest reliability ($n = 27$, $45 \pm 21$ years). Participants completed the LTPAQ-D, other self-report measures of aerobic and strength training PA, as well as tests of cardiorespiratory fitness (i.e., peak oxygen consumption ($\dot{V}O_{2peak}$)) and muscular strength. LTPAQ-D measures of min/week of aerobic LTPA, aerobic moderate-to-vigorous PA (MVPA), and strength training shared medium-to-large correlations with other self-report measures of aerobic and strength training PA ($r = 0.458–0.942$, $ps < 0.01$). After controlling for age, aerobic LTPA and MVPA shared moderate partial correlations with $\dot{V}O_{2peak}$ ($r = 0.341$ and $0.356$, respectively). Min/week of strength training, measured by the LTPAQ-D, was associated with predicted maximal strength on the chest press ($r = 0.621$, $p = 0.009$). All LTPAQ-D measures demonstrated good-to-excellent test–retest reliability (intraclass correlations $= 0.709–0.948$, $ps < 0.01$). This study provides preliminary evidence of the validity and reliability of the LTPAQ-D as a measure of LTPA among people with disabilities.

**Keywords:** physical activity; exercise; measurement; disability; impairment

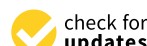



## 1. Introduction

The World Health Organization (WHO) developed physical activity (PA) guidelines for people living with disabilities that recommend (a) a minimum of 150 min/week of moderate, 75 min/week of vigorous, or an equivalent combination of moderate-to-vigorous-intensity PA (MVPA); and (b) muscle strength training activities involving all major muscle groups on two or more days/week at a moderate or greater intensity [1]. Because there is so little epidemiological data on the association between PA and health in people with disabilities [2,3], these guidelines were based almost entirely on studies of people without disabilities [4]. In order to collect much-needed data on PA and its relation to health among individuals with disabilities, valid and reliable measures of PA are urgently needed [2,5].

Self-report questionnaires are generally considered to be feasible measures of PA in large epidemiological studies, as they are typically low-cost, time-efficient, and easy to administer [5,6]. A number of self-report PA measures have been developed for people

with disabilities to capture the types of activities and mobility patterns of people with disabilities, which may differ from the general population; however, each has its limitations. For example, the Physical Activity and Disability Survey (PADS) does not measure exercise intensity, which is an essential element of the PA guidelines. The Physical Activity Scale for Individuals with Physical Disabilities (PASIPD) uses categorical response options to measure time spent on PA, which limits the specificity and accuracy of estimates of weekly minutes of PA [7]. Also, the PASIPD measures PA in metabolic equivalents (METs) based on data from individuals without disabilities [8], which may not be valid in disability populations [7]. In addition, no self-report measure assesses all four components of the WHO PA guidelines (i.e., frequency, intensity, duration, type), and therefore cannot assess whether respondents are meeting the guidelines. These limitations have led to a call for improved, standardized measures of PA for people with disabilities that can be used in PA surveys and surveillance systems [2].

The Leisure Time Physical Activity Questionnaire for People with Spinal Cord Injury (LTPAQ-SCI) is a self-report measure of the frequency (day/week), intensity (mild, moderate, vigorous), duration (min/week), and type of LTPA (i.e., PA individuals choose to engage in in their free time) performed over the past seven days [9]. Because it captures all components of the WHO PA guidelines, the questionnaire may also be used to determine whether respondents are meeting these guidelines.

The purpose of this study was to develop a version of the LTPAQ-SCI that can be used in studies including adults with physical and/or sensory disabilities. Our first objective was to adapt the LTPAQ-SCI into the LTPAQ for People with Disabilities (i.e., the LTPAQ-D) and to confirm its content validity through interviews with people living with disabilities. Our second objective was to evaluate the psychometric properties of the LTPAQ-D through assessments of its construct validity and test–retest reliability in a sample of people living with physical and/or sensory disabilities.

We hypothesized that min/week of LTPA and MVPA measured by the LTPAQ-D would be positively associated with: (a) LTPA participation measured by other PA questionnaires and (b) measures of cardiovascular fitness and muscular strength, which are outcomes known to increase in response to LTPA in people with disabilities [2]. We further hypothesized that LTPAQ-D measures of aerobic and strength training activity would demonstrate good-to-excellent test–retest reliability (i.e., intraclass correlation (ICC) > 0.75).

## 2. Materials and Methods

All participants gave their informed consent before they participated in the study. The study was approved by the University of British Columbia's Behavioural Research Ethics Board (H22-00250) and conformed to The Declaration of Helsinki. The protocol was guided by Terwee et al.'s [10] checklists for appraising the attributes and measurement properties of PA questionnaires.

### 2.1. Development and Content Validation

Five individuals who identified as having a physical and/or sensory disability were purposively sampled for interviews about the applicability of a recently revised version of the LTPAQ-SCI to other disability populations. The participants (2M/3F) represented various activity levels (i.e., sedentary to elite athlete) and lived with disabilities including cerebral palsy, multiple amputations, chronic pain, hearing impairment, and visual impairment. Participants were emailed the LTPAQ-SCI prior to their interview, which was conducted and recorded via videoconference (Zoom Video Communications, San Jose, CA, USA). During interviews, participants completed the questionnaire with a member of the research team, and then answered twelve questions about the content and coverage of LTPAQ-SCI items as measures of LTPA for people with disabilities (i.e., content validity) [11]. Following the initial round of five interviews, members of the research team discussed the responses and determined that we had reached saturation in the responses to

the interview questions. The participants were satisfied with the LTPAQ-D items, affirming content validity.

## 2.2. Construct Validation and Reliability

A cross-sectional study design was used to test the construct validity of the LTPAQ-D. Construct validation assessed whether the construct of LTPA participation, as measured by the LTPAQ-D, operated predictably within a system of measures of PA participation, cardiorespiratory fitness, and muscular strength.

Test–retest reliability was assessed by administering the LTPAQ-D to participants in the morning and evening of the same day. This was performed so that each administration of the LTPAQ-D covered the same recall period.

## 2.3. PA Questionnaire Measures

### 2.3.1. Leisure Time Physical Activity Questionnaire for People with Disabilities

The LTPAQ-D measures the number of days and minutes one is engaged in mild-, moderate-, and vigorous-intensity aerobic LTPA, as well as the number of days and minutes engaged in strength training LTPA, over the previous seven days. In addition, we summed the total time engaged in aerobic LTPA of any intensity as well as the time engaged in MVPA (i.e., moderate + vigorous LTPA). Time engaged in strength training was calculated by subtracting the average min/day spent resting during strength training LTPA from the average min/day spent on strength training multiplied by the number of days of strength training. All LTPA was calculated as combined aerobic and strength training LTPA. Definitions of aerobic and strength training LTPA and the different intensities of aerobic LTPA were provided to participants, and the questionnaire could be completed in under five minutes.

### 2.3.2. Physical Activity Scale for Individuals with Physical Disabilities (PASIPD)

The PASIPD was developed to assess PA in individuals with visual, auditory, or locomotor disabilities [12]. The PASIPD includes 13 items to measure the duration and frequency of occupational, household, and leisure activities performed over the past seven days. The questionnaire uses categorical response options, and each item has a multiplier value reflecting the activity's intensity in MET h/day. The total PASIPD score is calculated by summing all MET h/day values. The questionnaire has demonstrated some evidence of construct validity and test–retest reliability in adults with disabilities [13].

### 2.3.3. International Physical Activity Questionnaire—Short Form (IPAQ-SF)

The IPAQ-SF assesses the frequency, intensity, and time spent engaged in PA, as well as time spent sitting, over the seven days prior to completing the questionnaire. The IPAQ-SF is a valid and reliable questionnaire for assessing PA participation in individuals without disabilities [14], but its validity for people with disabilities is questionable [6,15,16]. Participants were asked to indicate on how many days they engaged in moderate PA, vigorous PA, and walking, and, if they indicated their participation in one or more of these activities, how many hours and minutes they spent engaged each day on average. Examples of activities and descriptions of feelings associated with moderate and vigorous intensity PA are provided to help respondents classify the intensity of their activities. Weekly participation in each type of PA was calculated as the number of days they engaged in the activity multiplied by the average time spent engaged on each of those days.

### 2.3.4. Physical Activity and Disability Survey

The PADS consists of four empirically derived subscales: exercise, LTPA, household activity, and time indoors [17]. For the purposes of the current study, only the exercise and LTPA subscales were administered to participants, as they reflect LTPA participation rather than the more general measure of PA. Participants were asked to indicate whether they participate in any exercise and/or LTPA and if so, the type, time, and frequency of

each activity per week. The total time in minutes spent engaging in exercise and LTPA was calculated by multiplying the total duration of each activity per day by the frequency in days. The PADS score was calculated by summing the exercise and LTPA score in min/week (i.e., PADS Exercise + LTPA). The survey has previously been demonstrated to be a reliable and valid tool for the assessment of PA levels in individuals with neurological conditions [18] and other chronic health conditions [17].

2.3.5. Behavioral Risk Factor Surveillance System Strength Training Questionnaire (BRFSS-SQ)

The BRFSS-SQ has been used to assess strength training participation in individuals without disabilities [19]. The questionnaire asks respondents to report whether they participate in any activities designed to increase muscle strength or tone in a usual week and, if so, how many days per week do they participate in such resistance-type activities.

*2.4. Measures of Muscle Strength and Cardiovascular Fitness*

Muscle strength was operationalized as the predicted one-repetition maximum (1RM; i.e., the maximum amount of weight a person can lift one time) and measured for two upper-body exercises, the chest press and seated row (Ab HUR Oy, Kokkola, Finland). We followed the protocol of Ribeiro Neto et al. [20], whereby participants completed as many repetitions as possible at a resistance at which they estimated they could complete a maximum of four to twelve repetitions. Resistance (in kilograms) and number of repetitions completed were entered into the following formula: Predicted 1RM = 1.942 + (1.102 × resistance) + (0.414 × repetitions).

Grip strength was assessed by a well-established protocol that has been described elsewhere [21]. In the seated position, participants gripped a hand dynamometer (Baseline Smedley Spring Dynamometer, Yagami Inc., Naka-ku, Nagoya, Japan) and exerted maximal effort for three trials lasting 3–5 s each, with one minute of rest in between trials. The protocol was completed on both the dominant and non-dominant hand. The highest value, reported in kilograms, from each hand was recorded.

A graded exercise test to exhaustion assessed peak oxygen consumption ($\dot{V}O_{2peak}$), a surrogate measure of cardiorespiratory fitness. Tests were administered by trained assessors and performed on either an electronically braked arm (Angio CPET arm ergometer; Lode, Groningen, The Netherlands) or leg (Excalibur sport leg ergometer; Lode, Groningen, The Netherlands) ergometer to accommodate differing participant abilities. Following two minutes of rest, both the arm and leg protocol began with participants cycling at 0 W. Resistance was then increased by 10 or 20 W each minute for the arm and leg protocol, respectively, until volitional fatigue. Participants were required to maintain a cadence of 55–65 rpm for arm and ~70 rpm for leg cycling. Breath-by-breath cardiopulmonary measures were recorded throughout the test (Cosmed K5, Albano Laziale, Italy), and the $\dot{V}O_{2peak}$ reported is the peak rolling 30 s average sampled at 10 s increments [22]. Peak power output ($PO_{peak}$) was recorded as the resistance (W) of the final stage completed.

*2.5. Procedure*

Sixteen participants who identified as having a physical and/or sensory disability and were able to travel to our laboratory completed the questionnaires and fitness tests on the same day. A schematic of the day's testing procedures is outlined in Figure 1. Participants unable to travel to the laboratory only completed the PA questionnaires. The questionnaires were completed in a randomized order and administered via face-to-face interviews by trained members of the research team, or by telephone or videoconference (Zoom Video Communications, San Jose, CA, USA) for those unable to travel to the laboratory. In the evening of the same day, the LTPAQ-D was completed again via telephone or videoconference interview for all participants (*n* = 27).

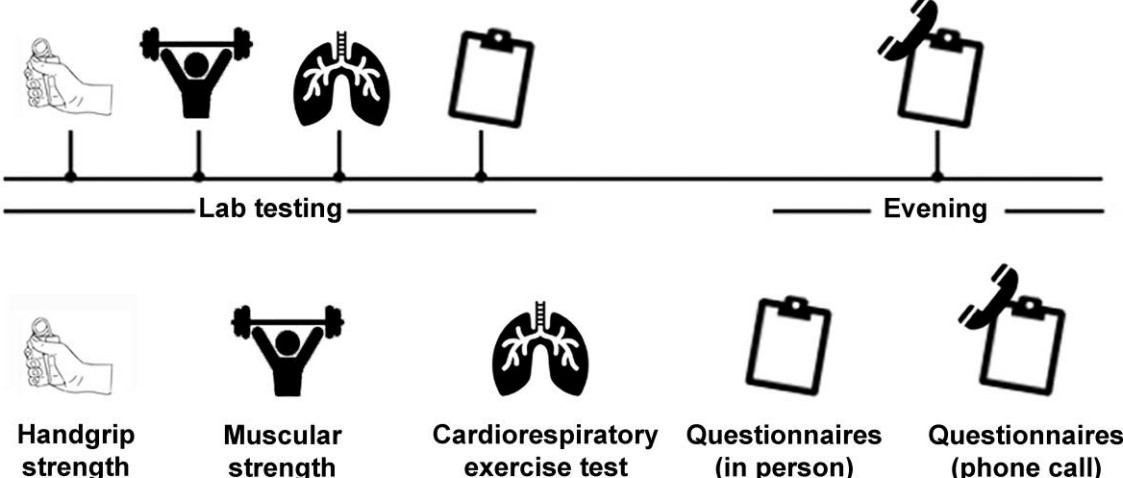

**Figure 1.** Schematic of study procedures for participants able to visit the laboratory.

*2.6. Sample Size Calculation*

Based on data collected during the development of the LTPAQ-SCI[R], we anticipated a large correlation ($r > 0.50$) between the LTPAQ-D measure of min/week of aerobic LTPA and the PASIPD score. We chose the PASIPD for sample size estimation because it has been frequently used to measure PA in studies involving people with disabilities and is considered a reasonable measure of PA in this population. Using a one-tailed Pearson correlation, a sample size of 21 was required to achieve statistical significance for $r = 0.50$, $\alpha = 0.05$, and 80% power.

*2.7. Statistical Analysis*

Analyses were conducted using IBM SPSS Statistics (v.29.0 for Mac, Armonk, NY, USA). Questionnaire data were checked for outliers, and values >3.29 standard deviations from the mean were scaled back to the next furthest value from the mean that remained within the normal range [23]. Measures of cardiorespiratory fitness and muscular strength were separately converted to Z-scores for participants who completed the arm and leg protocol. Each variable was tested for normality using a Shapiro–Wilk test. Square-root transformations were computed on all non-normal variables. All subsequent analyses used square-root transformations of LTPAQ-D variables.

One-tailed Pearson correlation coefficients between LTPAQ-D variables and other PA questionnaire measures were computed. As age is negatively related to fitness and strength [24,25], one-tailed partial correlations were performed between LTPAQ-D variables and measures of cardiorespiratory fitness and strength that controlled for age. Small, medium, and large correlations were interpreted as $r > 0.10$, $r > 0.30$, and $r > 0.50$, respectively [26].

ICCs were calculated from the two administrations of the LTPAQ-D using two-way random effects models to determine the test–retest reliability of the questionnaire. ICC values <0.5, 0.5–0.75, 0.75–0.9, and >0.90 were considered poor, moderate, good, and excellent, respectively [27]. Significance was set at $p < 0.05$.

## 3. Results

*3.1. Participant Characteristics*

Twenty-seven adults ($45 \pm 21$ years, 52% male) living with a physical and/or sensory disability were recruited from the local community between August 2022 and July 2023 through an established network and via social media advertising. Participants had impairments including cerebral palsy, multiple sclerosis, chronic pain, osteoarthritis, hearing impairment, and visual impairment, and reported living with a disability for $14 \pm 8$ years (Table 1). Participants reported a range of LTPA participation—44% of participants reported engaging in >150 min/week of MVPA as measured by the LTPAQ-D and 19% reported no

MVPA over the previous seven days. There were no differences in demographics among participants who did versus did not come to the laboratory for testing (all *ps* > 0.05).

**Table 1.** Participant demographics and LTPA participation as measured by the LTPAQ-D.

| Demographic Variables | *n* (% Total); Mean ± SD |
|---|---|
| **Sex** | |
| Male | 14 (52%) |
| Female | 13 (48%) |
| Age (years (range)) | 45 ± 21 (18–78) |
| Height (m) | 1.74 ± 0.11 |
| Male | 1.78 ± 0.09 |
| Female | 1.69 ± 0.11 |
| Body mass (kg) | 82 ± 24 |
| Male | 84 ± 18 |
| Female | 79 ± 29 |
| Years living with disability | 14 ± 8 |
| LTPAQ-D measures | |
| Mild-intensity aerobic (min/week) | 236 ± 202 |
| Moderate-intensity aerobic (min/week) | 104 ± 114 |
| Vigorous-intensity aerobic (min/week) | 50 ± 94 |
| Aerobic LTPA (min/week) | 391 ± 300 |
| Aerobic MVPA (min/week) | 154 ± 157 |
| Strength training (day/week) | 2 ± 2 |
| Strength training (min/week) | 92 ± 169 |
| All LTPA (min/week) | 483 ± 398 |

Abbreviations: LTPA, leisure time physical activity; LTPAQ-D, Leisure Time Physical Activity Questionnaire for People with Disabilities; MVPA, moderate-to-vigorous physical activity.

### 3.2. Construct Validity

#### 3.2.1. Aerobic LTPA

LTPAQ-D measures of aerobic LTPA and MVPA participation were positively and significantly correlated with all other questionnaire measures of LTPA and MVPA (*r* = 0.458 to 0.744, all *p* < 0.01; Table 2). Both min/week of aerobic LTPA and aerobic MVPA shared medium-sized partial correlations with $\dot{V}O_{2peak}$ (*r* = 0.341 and 0.356, respectively); however, these did not reach significance (*ps* ≥ 0.088).

**Table 2.** Correlations between LTPAQ-D measures of aerobic LTPA, aerobic MVPA, and all LTPA with other questionnaire measures of LTPA and cardiorespiratory fitness.

| Measure | LTPAQ-D Aerobic LTPA (min/week) | LTPAQ-D Aerobic MVPA (min/week) | LTPAQ-D All LTPA (min/week) |
|---|---|---|---|
| PASIPD (MET h/day) | 0.662 [†] | 0.577 [*] | 0.613 [†] |
| IPAQ-SF LTPA (min/week) | 0.635 [†] | 0.458 [*] | 0.657 [†] |
| IPAQ-SF MVPA (min/week) | 0.651 [†] | 0.649 [†] | 0.656 [†] |
| PADS Exercise + LTPA (min/week) | 0.744 [†] | 0.536 [*] | 0.718 [†] |
| $\dot{V}O_{2peak}$ (mL/kg/min) | 0.341 | 0.356 | 0.387 |

[*] indicates *p* < 0.01 (one-tailed); [†] indicates *p* < 0.001 (one-tailed). Correlations with $\dot{V}O_{2peak}$ are partial correlations. For LTPAQ-D: Aerobic LTPA = min/week of mild + moderate + vigorous intensity LTPA. Aerobic MVPA = min/week of moderate + vigorous intensity LTPA. All LTPA = min/week of aerobic LTPA+ min/week of strength training LTPA. Abbreviations: LTPA, leisure time physical activity; LTPAQ-D Leisure Time Physical Activity Questionnaire for People with Disabilities; MET, metabolic equivalent; MVPA, moderate to vigorous leisure time intensity physical activity; IPAQ-SF, International Physical Activity Questionnaire—Short Form; PADS, Physical Activity Disability Survey; PASIPD, Physical Activity Scale for Individuals with Physical Disabilities; $\dot{V}O_{2peak}$, peak oxygen consumption.

### 3.2.2. Strength Training LTPA

LTPAQ-D measures of strength training, both in day/week and min/week, shared large correlations with the BRFSS-SQ strength training questionnaire ($rs \geq 0.861$, $ps < 0.001$; Table 3), but were not significantly correlated with $PO_{peak}$ or measures of muscular strength (see Table 4). After controlling for age, strength training LTPA shared large, positive, and significant partial correlations with predicted 1RM on the chest press ($rs \geq 0.514$, $ps < 0.01$). Measures of strength training LTPA shared medium-sized ($r = 0.254–0.429$), albeit not significant, partial correlations with predicted 1RM on the seated row ($ps \geq 0.063$).

**Table 3.** Correlations between LTPAQ-D measures of strength training LTPA, the BRFSS-SQ measure of strength training, and strength tests.

| Measure | LTPAQ-D Strength Training (day/week) | LTPAQ-D Strength Training (min/week) |
|---|---|---|
| BRFSS-SQ (day/week) | 0.942 [†] | 0.861 [†] |
| $PO_{peak}$ (Watts) | 0.182 | 0.201 |
| Seated Row Predicted 1RM (kg) | 0.254 | 0.429 |
| Chest Press Predicted 1RM (kg) | 0.514 * | 0.621 * |
| Grip Strength Dominant (kg) | 0.351 | 0.390 |
| Grip Strength Non-Dominant (kg) | 0.065 | 0.240 |

* indicates $p < 0.01$ (one-tailed); [†] indicates $p < 0.001$ (one-tailed). Correlations with non-questionnaire-based measures are partial correlations. Abbreviations: 1RM, one-repetition maximum; BRFSS-SQ, behavioral risk factor surveillance system strength training questionnaire; LTPAQ-D, Leisure Time Physical Activity Questionnaire for People with Disabilities; $PO_{peak}$, peak power output.

**Table 4.** Test–retest reliability of the revised Leisure Time Physical Activity Questionnaire for People with Disabilities.

| Measure | Intraclass Correlation (95% CI) | *p*-Value |
|---|---|---|
| *Measures of aerobic LTPA* | | |
| Mild-intensity aerobic (min/week) | 0.709 (0.454–0.857) | <0.001 |
| Moderate-intensity aerobic (min/week) | 0.885 (0.764–0.946) | <0.001 |
| Vigorous-intensity aerobic (min/week) | 0.814 (0.681–0.924) | <0.001 |
| Aerobic LTPA (min/week) | 0.792 (0.592–0.899) | <0.001 |
| Aerobic MVPA (min/week) | 0.923 (0.838–0.964) | <0.001 |
| *Strength training LTPA* | | |
| Strength training (day/week) | 0.908 (0.808–0.957) | <0.001 |
| [a] Strength training (min/week) | 0.836 (0.675–0.921) | <0.001 |
| [b] Strength training total time (min/week) | 0.948 (0.890–0.976) | <0.001 |
| [c] Strength training resting time (min/week) | 0.775 (0.568–0.891) | <0.001 |
| *Combined aerobic + strength training LTPA* | | |
| All LTPA (min/week) | 0.751 (0.522–0.878) | <0.001 |

[a] Strength training (min/week) is calculated as: [average min/day strength training total time minus average min/day strength training resting time] × strength training day/week. [b] Strength training total time (min/week) is total time spent strength training, including the [c] time spent taking breaks in between sets of exercises. Statistical software did not compute a *p*-value for strength training (day/week) as test–retest data were an exact match. Abbreviations: CI, confidence interval; LTPA, leisure time physical activity; MVPA, moderate-to-vigorous intensity leisure time physical activity.

### 3.2.3. Combined Aerobic and Strength Training LTPA

All LTPA, measured by the LTPAQ-D in min/week (i.e., aerobic + strength training LTPA), shared large, positive, and significant correlations with all other questionnaire measures of LTPA and MVPA participation ($ps < 0.001$; Table 2). After controlling for age, all LTPA had a medium-sized positive, albeit not significant, partial correlation with $\dot{V}O_{2peak}$ ($r = 0.387$, $p = 0.069$).

*3.3. Test–Retest Reliability*

All variables measured by the LTPAQ-D exhibited good-to-excellent test–retest reliability (*ps* < 0.001; Table 3). ICCs for measures of aerobic LTPA ranged from 0.709 to 0.923; for strength training LTPA, they ranged from 0.775 to 0.948; and for all LTPA, the ICC was equal to 0.751.

## 4. Discussion

Valid and reliable measures of PA are necessary for use among people with disabilities [2]. The LTPAQ-D is an easy-to-administer, minimally burdensome questionnaire that assesses the frequency, intensity, and duration of aerobic LTPA and the frequency and duration of strength training LTPA. The LTPAQ-D aligns with current PA guidelines for people with disabilities and may be used to assess the effectiveness of interventions or programs that aim to enhance LTPA among people with disabilities.

We conducted an assessment of the content validity, as well as a preliminary assessment of the construct validity and test–retest reliability of the LTPAQ-D in a sample of adults with physical and/or sensory disabilities. User interviews supported the content, applicability, and layout of the questionnaire. In support of our hypotheses, individual and composite variables collected by the LTPAQ-D demonstrated good-to-excellent test–retest reliability.

Also, as hypothesized, LTPAQ-D measures of min/week of aerobic LTPA, aerobic MVPA, and all LTPA shared large correlations with all other self-reported measures of PA for people with disabilities. Associations between LTPAQ-D composite measures of LTPA shared medium-sized correlations with cardiovascular fitness (i.e., $\dot{V}O_{2peak}$). Despite correlations with cardiovascular fitness not reaching significance, the sizes of the correlations (*r* = 0.341 to 0.387) were similar to correlations previously reported between $\dot{V}O_{2peak}$ and other self-report PA measures for people with and without disabilities. For example, a systematic review of 23 IPAQ-SF construct validation studies reported a median *r* of 0.300 between self-reported LTPA and cardiorespiratory fitness in people with and without disabilities [28]. When considered in the context of other well-used self-report measures of LTPA, these findings support the validity of the LTPAQ-D as a measure of aerobic LTPA, MVPA, and all LTPA in people with physical and/or sensory disabilities.

Strength training LTPA, measured in days/week and min/week, shared large, positive, and significant associations with the BRFSS-SQ measure of strength training activities, which is consistent with our hypothesis. In partial support of our hypotheses, measures of strength training had large positive correlations with predicted 1RM on the chest press, but not the seated row exercise. While the explanation for this discrepancy is not clear, we suggest that other factors besides participation in strength training may influence 1RM—such as mode of mobility (e.g., wheelchair vs. ambulatory), type of impairment, and/or activities of daily living—and therefore undermine the strength of the relationships. Given the lack of validated measures of strength training PA for those with or without disabilities, we encourage researchers to explore other methods of measuring strength training participation that may be used to further validate the LTPAQ-D measures of strength training LTPA.

*Study Limitations*

Combining data from people with different physical and/or sensory disabilities may be considered a limitation, given that individuals with different impairments do not necessarily respond similarly to the same amount of exercise, creating variability in the correlations between PA and fitness. We have previously cautioned against grouping individuals with a wide range of impairments together [29,30], but took this approach in our study in order to maximize statistical power and relevance to the WHO guidelines for people with disabilities in general. Statistical standardization controlled for some, but not all, of the variability in impairment and its influence on fitness measures. We remind the reader that the purpose of the present study was to perform a preliminary evaluation of

LTPAQ-D measurement properties, and that additional construct validation studies are needed to test the validity of the LTPAQ-D for specific disability groups.

The study participants reported more min/week of LTPA than participants in other studies of adults with disabilities, where up to 50% have reported no LTPA whatsoever [2]. As such, this sample provided a more normal distribution of LTPA values and increased the likelihood that we would detect significant correlations [31]. However, we recognize that the sample may not be representative of the larger population of adults with disabilities, who are up to 62% less likely to meet PA guidelines than the general population [2].

The use of accelerometers may have provided another measure for testing the construct validity of the LTPAQ-D. We chose not to use accelerometers in the present study because they cannot distinguish between LTPA and other types of PA, and therefore do not measure the same behavior as the LTPAQ-D. In addition, data from individuals without disabilities suggest that accelerometers have issues with sensitivity to certain movements [32], and this may be an even greater issue if individuals have abnormal gait patterns. Further, in the context of the present study, accelerometers are not considered suitable tools for measuring PA among wheelchair users or whether they are meeting MVPA guidelines [7], as they do not accurately measure all wheelchair-based activity or strength training [33].

Finally, the process of measurement validation is ongoing, as no single study can definitively "validate" a measure [11]. The results of this study may be used only to substantiate inferences based on LTPAQ-D scores and the reliability and validity of the LTPAQ-D for use in a particular context and sample. While we expect that the LTPAQ-D would demonstrate validity and reliability when administered in other settings, and to people with specific types of impairments and disabilities, these hypotheses need to be tested. Researchers who wish to use the LTPAQ-D to measure LTPA in individuals with specific impairments or disabilities may first need to validate the questionnaire within these populations.

## 5. Conclusions

The development of the LTPAQ-D addresses the urgent need for an easy-to-administer, valid, and reliable measure of LTPA for people with disabilities. This study provides preliminary evidence of its construct validity and reliability among individuals living with physical and/or sensory disabilities. Collecting data on LTPA participation using a valid and reliable measure is essential to advance the knowledge regarding the influence of PA on health outcomes in people with disabilities, and to continue to develop and refine PA guidelines [5,34]. We encourage researchers to begin using the LTPAQ-D to assess LTPA in studies involving people with physical and sensory disabilities.

**Author Contributions:** C.M.G. acquired data, interpreted results, and drafted the manuscript for important intellectual content. A.N. acquired data and drafted the manuscript. A.J. acquired data. K.O. acquired data. K.A.M.G. conceived of and designed the work that led to the submission, interpreted results, and revised the manuscript for important intellectual content. All authors have read and agreed to the published version of the manuscript.

**Funding:** Data collection was supported by the Canadian Disability Participation Project, which is funded by the Social Sciences and Humanities Research Council of Canada (grant no. 895-2013-1021). KAMG holds the Reichwald Family Southern Medical Program Chair in Preventive Medicine.

**Institutional Review Board Statement:** The study was conducted in accordance with the Declaration of Helsinki. The Research Ethics Board of the University of British Columbia gave approval for this research (H22-00250, approved 5 May 2022).

**Informed Consent Statement:** Informed consent was obtained from all subjects involved in the study.

**Data Availability Statement:** The data that support the findings of this study are available upon request from the corresponding author.

**Conflicts of Interest:** The authors declare no conflicts of interest.

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
