# Peer review of "The Leisure Time Physical Activity Questionnaire for People with Disabilities: Validity and Reliability"

_disabilities, doi:10.3390/disabilities4020021_

Round 1
Reviewer 1 Report
Comments and Suggestions for Authors
This study addresses a critical need for questionnaires and assessments for examining physical activity in individuals with disabilities. The introduction and background is robust including the most current literature on this topic. In the last paragraph of the introduction, I believe the purpose could be more succinct or the divided into two AIMS - one for development of the tool and a second examining psychometric property. The methods were complete and well written with appropriate citations for the instruments included. I appreciated the inclusion of calculation for sample size as it addressed the preliminary concern of the relatively small sample size. Results were clearly explained and benefited from the tables provided. Discussion was comprehensive and thoroughly addressed my only concern of such a broad definition and inclusion criteria for disabilities. This article is well written and provides a clinically relevant tool with strong psychometric properties examining physical activity in individuals with disabilities.
Author Response
Reviewer 1:
This study addresses a critical need for questionnaires and assessments for examining physical activity in individuals with disabilities. The introduction and background is robust including the most current literature on this topic. In the last paragraph of the introduction, I believe the purpose could be more succinct or the divided into two AIMS - one for development of the tool and a second examining psychometric property. The methods were complete and well written with appropriate citations for the instruments included. I appreciated the inclusion of calculation for sample size as it addressed the preliminary concern of the relatively small sample size. Results were clearly explained and benefited from the tables provided. Discussion was comprehensive and thoroughly addressed my only concern of such a broad definition and inclusion criteria for disabilities. This article is well written and provides a clinically relevant tool with strong psychometric properties examining physical activity in individuals with disabilities.
We thank the reviewer for their time to review our manuscript and appreciate the feedback provided.
As per the reviewer’s suggestion we have revised the purpose/aims in the last paragraph of the introduction by separating it into two aims. In addition, we have moved the hypotheses to a separate paragraph. The section on the study purpose now reads:
“The purpose of this study was to develop a version of the LTPAQ-SCI that can be used in studies including adults with physical and/or sensory disabilities. First. this involved adapting the LTPAQ-SCI into the LTPAQ for People with Disabilities (i.e., the LTPAQ-D) and then performing a preliminary evaluation of its content validity through interviews with people living with disabilities. Second, we aimed to evaluate the psychometric properties of the questionnaire through assessments of its construct validity, and test-retest reliability of the newly developed LTPAQ-D in people living with disabilities.”
This change is highlighted in the updated version of the manuscript.
Reviewer 2 Report
Comments and Suggestions for Authors
Nice work!
Consider a good description of LTPAQ that is outside of the LTPAQ-D description.
“A Canadian-French version of the LTPAQ has demonstrated good-to-excellent test-restest 111 reliability in people with physical disbilities.” I think this needs to be mentioned earlier in manuscript and disabilities in the sentence is mis-spelled.
"Total PADS score was calculated by summing the LTPA score and exercise 140
score in min/week." How a total PADS score calculated if only 2 sections administered? And why only these 2 sections for this study?
Following two minutes rest,- add “of”
Author Response
Reviewer 2:
We thank the reviewer for their time to review the manuscript and appreciate the feedback provided. Responses to comments are provided point-by-point below.
Consider a good description of LTPAQ that is outside of the LTPAQ-D description.
We have expanded on our description of the LTPAQ in the introduction. Changes are highlighted in the revised manuscript. The description now states:
“The Leisure Time Physical Activity Questionnaire for People with Spinal Cord Injury (LTPAQ-SCI) is a self-report measure of the frequency (days/week), intensity (mild, moderate, vigorous), duration (min/week), and type of LTPA (i.e., PA individuals choose to do in their free time) performed over the past seven days [9]. Because it captures all components of the WHO PA guidelines, the questionnaire may also be used to determine if respondents are meeting these guidelines.”
Further detail on the questionnaire and how it is scored is provided within the LTPAQ-D description in the methods section.
“A Canadian-French version of the LTPAQ has demonstrated good-to-excellent test-retest reliability in people with physical disbilities.” I think this needs to be mentioned earlier in manuscript and disabilities in the sentence is mis-spelled.
Thank you for picking up on the spelling error. On reflection we have chosen to remove mention of this previous version of the LTPAQ as we feel it may confuse the reader. We used the Canadian-French version to develop a version of the LTPAQ for people with SCI from which this version (i.e., the LTPAQ-D) is based on.
"Total PADS score was calculated by summing the LTPA score and exercise score in min/week." How a total PADS score calculated if only 2 sections administered? And why only these 2 sections for this study?
We agree with the reviewer that only using two subscales from the PADS technically does not mean that this is a ‘Total’ score. In Table 2 we refer to this score as “PADS Exercise+LTPA” and have edited our description in the Methods to align with this.
Our reasoning for choosing to use only the exercise and LTPA subscales is that we wanted to use only the PADS subscales that assessed LTPA ((i.e., PA individuals choose to do in their free time) rather than all PA (i.e., any bodily movement produced by skeletal muscles that results in energy expenditure (Caspersen 1985)). We appreciated the reviewer for highlighting that this may not be clear as written in the manuscript and have taken this opportunity to add the following to our description of the PADS:
“For the purposes of the current study, only the exercise and LTPA subscales were administered to participants as they reflect LTPA participation rather than the more general measure of PA.”
This change has been highlighted in the manuscript.
Following two minutes rest,- add “of”
This has been corrected and highlighted in the revised manuscript. Thank you.
Reviewer 3 Report
Comments and Suggestions for Authors
Thank you for the opportunity to review this very interesting and well-prepared study. The study reports content and construct validity, and test-retest reliability of a self-reported physical activity questionnaire among people with disabilities. The topic of the study is highly relevant since there is an urgent need for valid and reliable measures of physical activity for people with physical and/or sensory disabilities. The results add new, encouraging information to be utilized in future studies.
However, there are several points that might be useful to be clarified:
1. Abstract: Please describe the number of participants in the validity and reliability analysis, indicate how this sample was drawn and a time-window for test-retest analysis.
2. Introduction: Please describe in brief how the ability to respond to self-reported PA-measures differs between people with disabilities the ones without. What kind of aspects needs to be considered?
3. Materials and methods: Please describe how was the study sample drawn? What was the age range and disability level of the participants? It is stated that “Sixteen participants able to travel to our laboratory completed the questionnaires and fitness tests on the same day.” Of whom were these able (n=16) and unable (n=9) participants selected? Were there any participants with more than one disability? How about cognitive disabilities? Were they excluded?
4. Materials and methods: It seems that BRFSS-SQ was not described in the methods. Please include.
5. Materials and methods: Did you consider use of accelerometers or other device-based measurements? Did you modify fitness tests somehow to be safe and feasible for the disabled participants?
6. Results, table 1: Data on height and weight would be more informative if it was presented separately for the males and females.
7. Results: Title 3.2. is Construct validity and below it there is Aerobic LTPA. Title 3.3. is Strength-training LTPA, which seems to include construct validity results as well. Please re-organize. Also, title 3.4. Combined aerobic and strength-training LTPA could be included under the title construct validity.
8. Results: No results for content validity are presented? It was mentioned only in chapter 2.1. Please describe the interview questions in brief.
9. Results/discussion: It seems that different disability types were combined in this study. I understand that with small number of participants it is not possible to conduct analysis according to disability groups, but this could be discussed in the discussion and presented as a need for future studies. It is plausible that physical and sensory disabilities have different effects on the participants ability to be physically active and to answer the questionnaires. This could be discussed in more detail.
10. Discussion: Please discuss about the potential use of device-based measurements when assessing validity of self-reported PA measures.
Minor correction
11. Introduction: One reference (Martin Ginis et al., 2021) has been written in full, although other ones are numbered. Please unify.
Author Response
Reviewer 3:
Thank you for the opportunity to review this very interesting and well-prepared study. The study reports content and construct validity, and test-retest reliability of a self-reported physical activity questionnaire among people with disabilities. The topic of the study is highly relevant since there is an urgent need for valid and reliable measures of physical activity for people with physical and/or sensory disabilities. The results add new, encouraging information to be utilized in future studies.
We thank the reviewer for their time to review the manuscript and appreciate the feedback provided which we believe have helped to strengthen the manuscript. We hope that with these edits the manuscript may not be acceptable for publication. Responses to comments are provided point-by-point below.
There are several points that might be useful to be clarified:
- Abstract: Please describe the number of participants in the validity and reliability analysis, indicate how this sample was drawn and a time-window for test-retest analysis.
To identify the number of participants in the validity and reliability analyses and to keep the abstract within the Journal’s abstract word limit we have replaced “2M/3F” with “n=5” and 14M/13F with “n=27”. We have also noted that test-retest reliability was performed within one day. All changes are highlighted.
- Introduction: Please describe in brief how the ability to respond to self-reported PA-measures differs between people with disabilities the ones without. What kind of aspects needs to be considered?
We have amended our sentence in the introduction regarding self-reported PA measures for people with disabilities to state:
“A number of self-report PA measures have been developed for people with disabilities to capture the types of activities and mobility patterns of people with disabilities which may differ from the general population; however, each has limitations.”
We hope that this addresses the reviewer’s question. We were a little unsure the meaning of the question - as this was the introduction was it not clear yet that individuals with cognitive disabilities were not included in this study? If we have not addressed appropriately we are happy to respond further with additional context.
- Materials and methods: Please describe how was the study sample drawn? What was the age range and disability level of the participants? It is stated that “Sixteen participants able to travel to our laboratory completed the questionnaires and fitness tests on the same day.” Of whom were these able (n=16) and unable (n=9) participants selected? Were there any participants with more than one disability? How about cognitive disabilities? Were they excluded?
We thank the reviewer for the opportunity to clarify our methods section. We hope that the changes described below make the studies methods clearer to readers.
Participants were recruited from the local communities between August 2022 and July 2023 through an established network and via social media advertising. We have edited and highlighted this within the ‘Participant Characteristics’ section.
The age range was 18-78. We have added this to Table 1 and highlighted.
Regarding the ‘disability level of the participants’ we purposefully chose not to classify the degree of impairment for each individual as we wish to first develop the LTPAQ-D as a questionnaire for the broader disability population. We acknowledge that this is a limitation of the present study.
We have included the following in the ‘Limitations’ section:
“While we expect the LTPAQ-D would demonstrate validity and reliability when administered in other settings, and to people with other types of impairments and disabilities, these hypotheses need to be tested.”
We have also tried to make it clearer that we included any individual who identified as having a physical and/or sensory disability within the description of the participants and these changes have been highlighted.
In terms of selecting participants who were and were not able to travel to the laboratory: this was not our decision, rather this was based on the individual circumstances of participants. We hope this clarifies the reviewer’s question.
Many participants had multiple impairments. For example, multiple sclerosis impacts many bodily functions and systems and has both physical AND sensory implications. As the focus of the present study was on the development and validation of the LTPAQ-D in the wider disability population we have refrained from trying to validate the questionnaire in specific sub-populations at this stage.
The reviewer is correct that individuals with cognitive disabilities were excluded from the present study. As stated in the introduction “The purpose of this study was to develop a version of the LTPAQ-SCI that can be used in studies including adults with physical and/or sensory disabilities”. We have been careful throughtout the results and discussion in stating that the findings of the study are specific to individuals with physical and/or sensory disabilities only.
- Materials and methods: It seems that BRFSS-SQ was not described in the methods. Please include.
Thank you for picking this up! The following description has been added:
“2.3.5. Behavioural Risk Factor Surveillance System Strength Training Questionnaire (BRFSS-SQ).
The BRFSS-SQ has previously been used to assess strength training participation in individuals without disabilities [20]. The questionnaire asks respondents to report whether they participated in any activities designed to increase muscle strength or tone in a usual week and, if so, how many days per week did they participate in such resistance-type activities.”
- Materials and methods: Did you consider use of accelerometers or other device-based measurements? Did you modify fitness tests somehow to be safe and feasible for the disabled participants?
Yes, we did consider the use of accelerometry in the present study. However, we opted against this assessment in the context of our study for a number of reasons. First, data from individuals without disabilities suggests that accelerometers have issues with sensitivity to certain movements and this may be an even greater issue if individuals have abnormal gait patterns (Sallis 2000), as is the case in many individuals with physical disabilities. Second, as our participants used a range of primary modes of mobility (i.e., ambulators and wheelchair-users) individualised cut-points to identify LTPA and the intensity thereof using specialised equipment (e.g., wheelchair treadmill) would need to be developed, and accelerometers are not considered suitable tools for measuring PA among wheelchair-users or whether they are meeting MVPA guidelines. Therefore, we did not consider this necessary or feasible for the present study. We have added the following to the ‘Limitations’ section of the manuscript:
“The use of accelerometers may have provided another measure for testing construct validity of the LTPAQ-D. We chose not to use accelerometers in the present study because they cannot distinguish between LTPA and other types of PA and therefore do not measure the same behaviour as the LTPAQ-D. In addition, data from individuals without disabilities suggests that accelerometers have issues with sensitivity to certain movements [33] and this may be an even greater issue if individuals have abnormal gait patterns. Further, in the context of the present study, accelerometers are not considered suitable tools for measuring PA among wheelchair-users or whether they are meeting MVPA guidelines [7] as they do not accurately measure all wheelchair-based activity and strength-training.[34]”
To ensure the safety of all individuals, exercise tests were performed by members of our laboratory who are trained on, and approved to, use of equipment. Training and approval were performed by a university-based certified exercise physiologist (i.e., CSEP-CEP) who is further credentialed to assess and certify exercise physiologists. We have noted and highlighted that tests were performed by a ‘trained assessor’. Participants were also given the option of performing the test using arm- or leg-ergometer modalities as a further safety measure. Protocols were developed by members of the research team who have extensive experience administering exercise tests with individuals with disabilities (and those without) and purposefully designed to start at a lower resistance and increase by smaller than usual increments to ensure the validity of testing and the feasibility of obtaining a true V̇O2peak value.
- Results, table 1: Data on height and weight would be more informative if it was presented separately for the males and females.
We have added height and weight separately for males and females in Table 1. These changes have been highlighted.
- Results: Title 3.2. is Construct validity and below it there is Aerobic LTPA. Title 3.3. is Strength-training LTPA, which seems to include construct validity results as well. Please re-organize. Also, title 3.4. Combined aerobic and strength-training LTPA could be included under the title construct validity.
Thanks for picking this up. We have re-organised as the reviewer has suggested so that it is now:
3.2. Construct Validity
3.2.1. Aerobic LTPA.
3.2.2. Strength-training LTPA.
3.2.3. Combined aerobic and strength-training LTPA.
3.3. Test-retest reliability
- Results: No results for content validity are presented? It was mentioned only in chapter 2.1. Please describe the interview questions in brief.
We chose to present these results within the methods under section ‘2.1 Development and Content Validation’ as we felt that this fit the format of the manuscript better and made the process of development through construct validation easier to follow for the reader. We have made small changes to this section to briefly describe the interview and process for content validation in more detail. This was a second stage of content validation and no changes were made – as noted, “Participants were satisfied with LTPAQ-D items, affirming content validity.”
- Results/discussion: It seems that different disability types were combined in this study. I understand that with small number of participants it is not possible to conduct analysis according to disability groups, but this could be discussed in the discussion and presented as a need for future studies. It is plausible that physical and sensory disabilities have different effects on the participants ability to be physically active and to answer the questionnaires. This could be discussed in more detail.
We agree that there is a need for validation in specific disability groups and that this is a preliminary study on the development of the questionnaire and its use in the general disability population. We have added the following to our discussion on the questionnaires use in different populations in our ‘Limitations’ section:
“Researchers who wish to use the LTPAQ-D to measure LTPA in individuals with specific impairments or disabilities may first need to validate the questionnaire within these populations.”
- Discussion: Please discuss about the potential use of device-based measurements when assessing validity of self-reported PA measures.
Thank you for the opportunity to address the use of device-based measurements. As mentioned above in response to question 5, we have added the following to the ‘Limitations’ section:
“The use of accelerometers may have provided another measure for testing construct validity of the LTPAQ-D. We chose not to use accelerometers because they cannot distinguish between LTPA and other types of PA and therefore do not measure the same behaviour as the LTPAQ-D. In addition, as data from individuals without disabilities suggests that accelerometers have issues with sensitivity to certain movements [33] and this may be an even greater issue if individuals have abnormal gait patterns. Further, in the context of the present study, accelerometers are not considered suitable tools for measuring PA among wheelchair-users or whether they are meeting MVPA guidelines [7] as they do not accurately measure all wheelchair-based activity and strength-training.[34]”
Minor correction
- Introduction: One reference (Martin Ginis et al., 2021) has been written in full, although other ones are numbered. Please unify.
Thank you for highlighting, this has been corrected and highlighted in the revised manuscript.